# Dynamics Changes of the Fecal Bacterial Community Fed Diets with Different Concentrate-to-Forage Ratios in Qinghai Yaks

**DOI:** 10.3390/ani12182334

**Published:** 2022-09-08

**Authors:** Kaiyue Pang, Yingkui Yang, Shatuo Chai, Yan Li, Xun Wang, Lu Sun, Zhanhong Cui, Shuxiang Wang, Shujie Liu

**Affiliations:** 1Qinghai Academy of Animal Husbandry, Veterinary Sciences in Qinghai University, Xining 810016, China; 2Key Laboratory of Plateau Grazing Animal Nutrition and Feed Science of Qinghai Province, Qinghai Academy of Animal Husbandry, Veterinary Sciences in Qinghai University, Xining 810016, China; 3Yak Engineering Technology Research Center of Qinghai Province, Xining 810016, China

**Keywords:** aks, forage-to-concentrate ratio, fecal, growth performance, microbiota

## Abstract

**Simple Summary:**

Yaks are important for the economy and livelihood of local herders in the Tibetan Plateau and are their primary source of income. Traditionally, the production performance of ruminants is generally improved by increasing the proportion of concentrate feed in the diet. To the best of our knowledge, the optimal proportion of concentrate feed is limited in actual production and research of yak. This study fed three diets with different dietary forage-to-concentrate ratios (50:50, 65:35, and 80:20) to 36 male yaks (*Bos grunniens*). The changes in the distribution of fecal microorganisms were studied by 16S rRNA gene sequencing and liquid chromatography–mass spectrometry. We found that the dietary forage-to-concentrate ratio significantly impacted yak fecal microorganisms. As such, these studies help us understand the fecal microbiota of yaks, further provide more complete information on the requirements of yak diets in terms of concentrate-to-forage ratio and improve yak growth performance, and provide a theoretical basis for accurate housing of yaks.

**Abstract:**

(1) Background: This study aimed to investigate the effects of different dietary concentrate to roughage ratios on growth performance and fecal microbiota composition of yaks by 16S rRNA gene sequencing. (2) Methods: In the present study, three diets with different dietary forage-to-concentrate ratios (50:50, 65:35, and 80:20) were fed to 36 housed male yaks. (3) Results: The result shows that Final BW, TWG, and ADG were higher in the C65 group than in the C50 and C80 groups, but the difference was not significant (*p* > 0.05). DMI in the C65 group was significantly higher than in the other two groups (*p* < 0.05). The DMI/ADG of the C65 group was lower than that of the other two groups, but the difference was insignificant (*p* > 0.05). At the phylum level, Firmicutes were the most abundant in the C65 group, and the relative abundance of Bacteroidetes was lower in the C65 group than in the other two groups. At the genus level, the relative abundances of *Ruminococcaceae_UCG_005*, *Romboutsia*, and *Christensenellaceae_R-7* were higher in the C56 group than in the C50 and C80 groups. The relative abundance of *Lachnospiraceae_NK3A20* and *Rikenellaceaewas_RC9_gut* is lower in the C65 group, but the difference was insignificant (*p* > 0.05). At KEGG level 2, the relative abundance of lipid metabolism and energy metabolism were lowest in the C50 group, and both showed higher relative abundance in the C65 group. (4) Conclusions: In conclusion, the structure of fecal microbiota was affected by different concentrate-to-forage ratios. We found that feeding diets with a concentrate-to-forage ratio of 65:35 improved yaks’ growth and energy metabolism.

## 1. Introduction

Yaks (*Bos grunniens*), an endemic livestock species living on the Qinghai–Tibetan Plateau (QTP), are an essential means of production and livelihood for local herders [1]. In recent years, the demand for animal products has been increasing. To meet the increasing demand for animal products in the market, increasing the proportion of concentrate in the diet to improve the production performance of ruminants is the most convenient and feasible. In the actual production process, the unreasonable diet structure will make the animal health and production levels decline; however, too high or too low a level of concentrate will affect the economic benefits of breeding, such as too high a level of concentrate will lead to a variety of nutritional, metabolic diseases in animals. At the same time, too low a level of concentration will cause slow growth of animals [2].

Presently, the response of the dietary structure to yak rumen microbiota has been a research hotspot in recent years [3,4], and there are many studies on yak rumen microbiota [5,6]. For example, Ahmad et al. [7] studied the effect of different dietary energy levels on the bacterial community in the yak rumen, and Zhang et al. [8] studied the effect of dietary protein levels on the dynamics of yak rumen microbiota. However, to our knowledge, the latest research on yak flora is mainly focused on the rumen, while little attention has been paid to the study of fecal microbiota. Tests on in vitro fermentation of rumen contents and fresh fecal showed that the digestibility of various feeds was positively correlated with the digestibility of feeds in yaks, suggesting that fecal microbiota can to some extent replace the function of rumen flora in animal food digestion by resolving flora [9]. Therefore, our study focused on the effect of the concentrate-to-forage ratio in the diet on the fecal microbiota of yaks.

The gastrointestinal tract of ruminants has a rich and diverse microbiota. It plays a crucial role in maintaining the digestive system’s homeostasis and the immune system’s function system [10]. Under normal circumstances, the microorganisms in the gastrointestinal tract are in a state of dynamic equilibrium with the host. Still, when the nutritional level of the diet changes greatly, it usually affects the structure of the flora in the gastrointestinal tract, thereby breaking this balance. Diet is a significant factor regulating microbial systems and metabolism in the gastrointestinal tract. Most of the gastrointestinal flora of mammals is concentrated in the hindgut, mainly in the colon. When comparing the structure of fecal flora and colon content flora, it is found that fecal flora can represent the intestinal content flora [11]. Therefore, it is important to understand the changes in the fecal microbiota of yaks-fed diets with different concentrate-to-forage ratios. Fecal microbiota can reflect the composition of the gut microbiota, and more and more researchers are studying the gut microbiota through fresh fecal microbiota composition [12,13,14]. 

We hypothesized that diets with different ratios of concentrate to roughage would affect the yak growth performance fecal the structure of microbiota. Therefore, the purpose of this study was to comprehensively study the effects of different concentrate-to-rough ratio diets on yak manure fermentation and microbiota to provide a theoretical basis for a yak to choose the appropriate diet concentrate-to-rough ratio. Further, 16S rRNA gene sequencing technology was used to study the structure of yak fecal microbiota flora and to explore the possible relationship between fecal fermentation parameters and microbial flora.

## 2. Materials and Methods

All experimental procedures and animal experiments were performed following the guidelines of the Ethics Committee. This study was approved by the Institutional Animal Care and Use Committee of Qinghai University (Protocol number: QHU20190918).

### 2.1. Animals, Diets, and Experimental Design

This study was conducted from September to December 2019 at the Laozaxi breeding in Guinan County, Qinghai Province, China. In total, the consistent initial weight of 36 3-year-old healthy male yaks (weight: 164.9 ± 12.9 kg) was selected from grazing pastures and randomly divided into three groups. The yaks in the three groups were alone fed diets with a 50:50 (C50), 65:35 (C65), or 80:20 (C80) concentrate-to-forage ratio. The diets were prepared according to the Chinese Beef Cattle Feeding Standard (NY/T815-2004), and the composition and nutrient content of the diets are shown in Table 1. All yaks were uniformly numbered, and each yak was kept separately in a pen with free access to water and fed twice a day at 08:00 and 17:00. The pretest period was 15 d, and the average test period was 90 d.

### 2.2. Forage Samples Collection and Measurements

Inedible forage was removed when collecting mixed forage samples, and only edible forage was retained. They were then stored in self-sealing bags at −20 °C for later analysis. Crude protein (CP), neutral detergent fiber (NDF), acid detergent fiber (ADF), calcium (Ca), and phosphorus (P) were determined in each sample in the laboratory, and ME was calculated. Mixed feeds (100 g) were collected and dried in a forced-air oven at 60°C for 48 h and then ground through a 1 mm sieve before analysis. CP, Ca, and P contents were determined according to AOAC Procedure (1990) [15]. The NDF and ADF contents were determined by the method of VanSoestd et al. [16].

### 2.3. Determination of Growth Performance and Fecal Samples Collection

At the beginning and end of the trial period, the test yaks were weighed on an empty stomach before feeding, and the average daily weight gain and total weight gain were calculated. The number of feeds and residuals were recorded daily, and dry matter intake and F/G were calculated.

At 3 to 4 h after feeding on the 90th day of the experiment. From the 36 yaks, six yaks were randomly selected from each of the three groups, and a total of 18 yaks were selected. About 50 g of fecal was collected by the rectal fecal extraction method. Fecal samples were quickly divided into 2 mL cryopreservation tubes and stored in liquid nitrogen to determine microbiota.

### 2.4. Microbial DNA Extraction, PCR Amplification, Sequencing, and Sequencing Data Processing

The CTAB method was used to extract microbial DNA from fecal samples. We used 1.0% agarose gel electrophoresis to detect the concentration and purity of DNA. The DNA was diluted to 1 ng/μL with sterile water according to the attention. The extracted DNA was amplified by PCR, using barcoded specific primers 515F (5′-GTGCCAGCMGCCGCGG-3′) and 806R (5′-GTGCCAGCMGCCGCGG-3′) to amplify different regions of the 16S rRNA gene (16S V3–V4). We used a 25 μL amplification system, five μmol/L upstream and downstream primers, and ~5 ng template DNA for the PCR reaction. PCR amplification conditions were as follows: 94 °C pre-denaturation treatment for 5 min, a denaturation cycle at 94 °C for 30 s, annealing at 50 °C for 30 s, and extension at 72 °C for 60 s for a total of 30 cycles; finally, a 72 °C extend for 7 min. The PCR amplification products were detected by 1.0% agarose gel electrophoresis; the recovered products were purified using the MinElute Gel Extraction Kit (Qiagen, Germany) and TruSeq^®^ DNA PCR-Free Sample Preparation Kit (Illumina, San Diego, CA, USA). Qubit and Q-PCR quantified the constructed library and then sequenced on the Illumina NovaSeq6000 platform. PCR amplification, PCR product mixing and purification, library construction, and computer sequencing processes were all produced by Allwegene Technology Co., Ltd. (Beijing, China) Technology Co., Ltd. (Beijing, China) to produce 250 bp paired-end readings.

The sequences obtained from the Illumina NovaSeq6000 platform were processed through the open-source software pipeline QIIME (Quantitative Insights into Microbial Ecology) version 1.8.0-dev [17], with the criteria as described by previous reports [18,19]. Briefly, (1) threads that had a mean quality score of no <20 and no shorter than 50 bp were retained; (2) we discarded reads that had exact barcode matching, two nucleotide mismatches in primer matching, and ambiguous characters; (3) only sequences that overlapped by more than ten bp were assembled according to their overlap sequence. Reads that could not be assembled were discarded. Sequences were binned into operational taxonomic units (OTUs) based on 97% identity using UCLUST (version7.1, http://drive5.com/uparse/, accessed on 8 July 2020), and chimeric sequences were identified and removed by UCHIME [20]. The most abundant sequence within each OTU from specific libraries (bacteria) was designated as the “representative sequence” and aligned against the SILVAbacterial database (version 119) [21], respectively, with the default parameters set by QIIME. Community richness and diversity were analyzed with measures such as Observed species, PD whole tree, Chao1 and Shannon indices, weighted uniFrac distance-based principal coordinate analysis (PCoA), and weighted distance-based analysis of molecular variance (AMOVA), which were used to illustrate significant differences among the samples, were assessed by the program MOTHUR v.1.35.0 [22]. 

### 2.5. Data and Statistical Analysis

Statistical analyses were performed using R (v 3.6.1). The statistical difference between normally distributed data was analyzed using a One-Way ANOVA and SPSS 26.0, and differences were considered statistically significant (*p* < 0.05). Linear discriminant analysis effect sizes (LEfSe, LDA > 3) were used to identify significant bacteria in both groups [23]. By spearman inspection method, select all samples before the absolute abundance of 20 genera level results correlation analysis, and using the corresponding door as the legend, calculation results to filter out the *p*-value is more significant than 0.05 or related value |R| < 0.6 for drawing. The prediction of rumen microbiota function in yaks fed different concentrate-to-forage ratios was studied with PICRUSt 2, and differences between the three groups were determined in level 2 of the KEGG (Kyoto Encyclopedia of Genes and Genomes) pathway [24].

## 3. Results

### 3.1. Effect of Dietary Concentrate-to-Forage Ratio on Growth Performance of Yaks

As shown in Table 2, Final BW, TWG, and ADG were the highest in C65 and the lowest in C50 compared to the other two groups, but the difference was insignificant (*p* > 0.05). In addition, DMI in the C65 group was significantly higher than in the other two groups (*p* < 0.05). The F/D of the C65 group was lower than that of the other two groups, but the difference was not significant (*p* > 0.05)

### 3.2. Changes in Bacterial Diversity of Yak Fecal in Three Treatment Groups

A total of 1,612,611 effective 16S rDNA gene sequences were obtained from eighteen yak fecal samples. After subsampling each sample to an equal sequencing depth and clustering, we obtained 2066 OTUs with a recognition rate of 97%. The 1342 OTUs were shared, accounting for 64.96% of the total number of OTUs. The number of OTUs in the C50, C65, and C80 groups were 1612, 1755, and 1787, respectively, with corresponding numbers of unique OTUs of 83, 105, and 132, respectively (Figure 1A). The unweighted PCoA analysis based on UniFrac showed that the contribution of the first and second principal components was 24.86% and 13.27%, respectively, and the bacterial structure of the fecal of each treatment group of yaks could be clearly separated (Figure 1B). The study showed that the structure of fecal colony structure was significantly different between diets with different concentrate-to-forage ratios, indicating that feeding different diets with different concentrate-to-forage ratios influenced the composition of fecal colony structure in yaks. The Chao 1 index measured the richness of the flora, the Shannon index and the coverage index of the PD-whole-tree measured the diversity of the flora, and Observed-species was used to analyze the number of OUTs, with higher indices indicating higher diversity and richness of the flora in the sample. By alpha diversity analysis (Figure 2), there were differences in the diversity and richness of the flora between the Chao1 value, the coverage index of PD-whole-tree, the Shannon index, and Observed-species.

### 3.3. Differences in the Bacterial Composition of Yak Fecal in the Three Treatment Groups

Based on the taxonomic analysis, a total of 17 bacterial phyla were identified from yak fecal samples fed the three types of refined to forage ratios. The dominant phylum was Firmicutes (76.11%), Bacteroidetes (11.99%), Actinobacteriota (5.85%), Proteobacteria (1.86%), and Spirochaetota (1.58%). Among them, Firmicutes and Bacteroidetes were the most abundant phylum in all samples, accounting for 76.11% and 11.99% of the total readings, respectively (Figure 3A). The other fewer accounted for phylum are Verrucomicrobiota (1.47%), Patescibacteria (0.64%), Cyanobacteria (0.40%), Fusobacteriota (0.04%) and Fibrobacterota (0.02%). At the genus level, a total of 235 genera of bacteria were identified. The main genera are as follows *Ruminococcaceae_UCG_005* (13.25%), *uncultured_bacterium* (8.52%), *Romboutsia* (6.31%), *Lachnospiraceae_NK3A20* (5.99%), *Christensenellaceae_R-7* (5.82%), *Rikenellaceae_RC9_gut* (3.74%), *Clostridium_sensu_stricto_1* (3.69%) (Figure 3B).

At the phylum level (Figure 3C), the relative abundance of Firmicutes was the most abundant in the C65 group, the lowest in the C50 group, the highest relative abundance of Bacteroidetes in the C80 group, and the lowest relative abundance in the C65 group, but the difference was not significant (*p* > 0.05). The relative abundance of Verrucomicrobiota was significantly higher in the C50 group than in the other two groups (*p* < 0.05). With the increase in dietary concentrate ratio, Firmicutes showed a trend of increasing and then decreasing, while Bacteroidetes showed a trend of decreasing and then increasing. At the genus level (Figure 3D), the relative abundances of *Ruminococcaceae_UCG_005*, *uncultured_bacterium*, *Romboutsia*, and *Christensenellaceae_R-7* were higher in the C56 group than in the C50 and C80 group. The relative abundance of *Ruminococcaceae_UCG_005*, *uncultured_bacterium*, *Romboutsia*, and *Christensenellaceae_R-7* all showed a trend of increasing and then decreasing with increasing the percentage of dietary concentrates. However, the relative abundance of *Lachnospiraceae_NK3A20* and *Rikenellaceaewas_RC9_gut* lower in the C65 group, the relative abundance of *Lachnospiraceae_NK3A20* was the highest in C50, and that of *Rikenellaceaewas_RC9_gut* was the highest in C80. Still, the difference was not significant (*p* > 0.05). Show that with the increase in the concentrate-to-forage ratio, the dietary concentrate-to-forage ratio decreased firstly and then increased.

### 3.4. LEfSe Analysis of the Bacterial Composition of Yak Fecal in Three Treatment Groups

We also performed LEfSe (Linear discriminant analysis Effect Size) to detect variations in the bacterial taxa composition. LDA values were greater than 3, and significant differences among the three concentrate-to-forage diets are shown in Figure 4. In addition, when microbial communities were compared in the context of different concentrate-to-forage ratio diets, the most abundant bacterial genera in the C50 group were *Prevotellaceae_UCG_001*, *Pygmaiobacter*, and *Tyzzerella.* At the same time, *Arthrobacter* and Arthrobacter_sp_Edens01 were C65 groups that were more abundant.

### 3.5. Analysis of Network Interaction between Different Genera

Using the Spearman test, the genus level results of the top 20 absolute abundances of all samples were selected for intercorrelation analysis (Figure 5). *Roseburia* was positively correlated with *Treponema* (*r* > 0.008, *p* < 0.01), *Acetitomaculum* was positively correlated with *Lachnospiraceae_NK3A20* (*r* > 0.81, *p* < 0.01) and negatively correlated with *Bacteroides* (*r* < −0.70, *p* < 0.01), *Atopobium* was positively correlated with *Olsenella* (*r* > 0.76, *p* < 0.01), *Romboutsia* was positively correlated with *Turicibacter* and *Paeniclostridiu* (*r* > 0.67, *p* < 0.01), *Paeniclostridium* was positively correlated with *Family_ XIII_AD3011* and *Romboutsia* were positively correlated (*r* > 0.69, *p* < 0.01), *Turicibacter* was positively correlated with *Clostridium_sensu_stricto_1* and *Romboutsia* (*r* > 0.61, *p* < 0.01).

### 3.6. Functional Prediction of Fecal Bacteria

To assess the functional characteristics of the fecal microbiota, we used PICRUSt2 to predict potential functions and compare the differences between the three dietary concentrates to forage ratios (Table 3). At KEGG level 2, the most abundant are carbohydrate metabolism, amino acid metabolism, metabolism of cofactors and vitamins, metabolism of terpenoids and polyketides, the relative abundance of metabolism of other amino acids, replication, and repair, lipid metabolism, energy metabolism was also high. The relative abundance of carbohydrate metabolism was highest in the C50 group and lowest in the C65 group, while the relative abundance of amino acid metabolism was highest in the C80 group and lowest in the C50 group. The relative abundance of lipid metabolism and energy metabolism were lowest in the C50 group, and both showed higher relative abundance in the C65 group.

## 4. Discussion

A suitable concentration-to-forage ratio can improve the growth performance and feed digestibility of ruminants [25]. We found that the ADG tended to increase in the C65 group, and it is possible that feeding a 65:35 concentrate-to-forage ratio would help improve yaks’ growth performance. DMI has a decisive role in the growth and development of ruminants, which is influenced by a variety of factors, including a greater relationship with the dietary concentrate-to-forage ratio [26]. The results of the present study showed that the DMI of yaks increased and then decreased with the increase in the dietary concentrate-to-forage ratio, which is consistent with the study of Johnson et al. (1985) [27], probably because feeding a 65:35 concentrate-to-forage ratio can meet the energy requirements of yaks for growth and development; however, too high concentrate ration can cause waste of feed resources. F/G is an essential indicator of animal growth performance, reflecting the ruminant’s ability to digest and utilize feed. The lower its value, the higher the feed conversion rate. In this study, C65 had the lowest F/G and the highest feed conversion ratio, indicating that yaks fed a 65:35 concentrate-to-forage ratio were receiving enough nutrients from the diet to meet their own needs and that the excess nutrients were used for growth and development. The more nutrients used for growth, the highest the feed conversion rate; this was also verified by feeding a higher ADG of 65:35 concentrate-to-forage.

This study used 16S rRNA gene sequencing technology to clarify the dynamic changes of the fecal microbiota of yaks-fed diets with different concentrate-to-forage ratios. As for β diversity, PCoA results showed that the composition and structure of bacteria in the fecal of the three groups were different, but it is not significant. The alpha diversity index indicates (including Chao1, Observed_species, PD_whole_tree, and Shannon) that additional concentrate-to-forage ratios affected the fecal microbiota richness and diversity of yaks. Unfortunately, there have been few studies on the effects of concentrate-to-forage ratio on bacterial communities in ruminant feces. Our study found that the diversity and structure of bacteria in fecal yaks fed diets with different concentrate-to-forage ratios were different.

In the present study, we identified Firmicutes and Bacteroidota as the predominant flora in yak fecal. Similar findings have been reported in previous studies [28,29], suggesting that Firmicutes and Bacteroidota play a vital role in the intestinal tract. Other studies have concluded that the dominance of Firmicutes and Bacteroidota may be due to changes in diet and climate [30]. The relative proportions of Firmicutes and Bacteroidota were different in the diets fed with a different ratio of concentrate-to-forage, which may be related to the function of bacteria. Firmicutes contain genes associated with energy metabolism and the breakdown of substances such as fiber and cellulose [31,32,33]. Bacteroidota has mainly biological functions of degrading proteins and carbohydrates [34,35,36]. According to our study, with the increase in dietary concentrate level, Firmicutes showed a tendency to rise and then fall. In contrast, Bacteroidota showed a tendency to fall and then rise, and the relative abundance of Firmicutes was higher. The relative abundance of Bacteroidota was lower in the C65 group, which shows the fecal microbiota of yaks fed a 65:35 concentrate-to-forage ratio. This indicates that the energy metabolism and material catabolism-related ability of yak fecal microbiota fed 65:35 concentrate to crude ratio is stronger. In our study, Actinobacteriota was the third most abundant phylum; despite its lower relative abundance compared to Firmicutes and Bacteroidota, it still plays a vital role in intestinal metabolism, for example, by participating in carbohydrate catabolism [37].

This study also examined the variation in genus-level microbiota. Mao et al. (2015) [17] found that the dominant bacteria in Holstein cow manure were unclassified *Ruminococcaceae*, *Peptostrepto-coccaceae*, and *Clostridium*, while Meale et al. (2017) [38] reported that unclassified *Ruminococcaceae* and *Bacteroides* were found to be the dominant bacteria in the fecal of weaned calves. Among the different genera represented in this study, *Ruminococcaceae_UCG_005*, *uncultured_bacterium*, and *Romboutsia* were the predominant bacteria in yak fecal, not similar to the above research, and the reason for the difference could be due to the breed, age, and diet of the animals. *Ruminococcaceae_UCG_005* has been shown to be involved in cellulose degradation and starch digestion in animals [39], and this species is presumed to play an essential function in the basal metabolism of yaks. *Romboutsia* is a short-chain fatty acid-producing bacterium known to be involved in glucose and oligofructose degradation [40,41]. Members of the *Christensenellaceae* family can secrete three glycosidases that promote the breakdown of cellulose and hemicellulose in forage fodder and improve forage utilization efficiency [42], and in the present study *Ruminococcaceae_UCG_005*, *Romboutsia* and *Christensenellaceae_R-7* all had the highest relative abundance in the C65 group, it is indicated that the ability of yaks fed 65:35 concentrate-to-forage ratio to degrade pasture cellulose and obtain energy from indigestible polysaccharides was improved to promote yak growth. *Lachnospiraceae* has been reported to affect body glucose levels by participating in energy metabolism [43] and is positively associated with obesity as a relevant physiological marker for the development of obesity [44]. *Rikenellaceae* is considered a gut microbial marker in the obese population [45], and this genus’s abundance has been significantly increased in and in high-fat mouse models [46,47]. In the present study, the relative abundance of *Lachnospiraceae_NK3A20* and *Rikenellaceae_RC9* were lower in the C65 group than in the other two groups, and feeding yaks at a 65:35 concentrate-to-forage ratio may result in a relatively low risk of developing metabolic disease.

Metabolites of gut microbes play an important role in maintaining their host physiology and metabolic homeostasis [48]. Predictions of rumen bacterial function in yaks suggest that yaks possess higher energy storage, lipid metabolism, glycan synthesis, and metabolic gene families than other breeds. Differences in these gene families may help yaks improve energy efficiency. In this study, we found that the dietary concentrate ratio influenced the gene function of yak fecal microbiota through PICRUSt2 gene function prediction. At the secondary level of the KEGG metabolic pathway, genes of yak fecal microbiota fed different dietary concentrate ratios were involved in metabolism-related pathways such as carbohydrate metabolism, amino acid metabolism, metabolism of cofactors and vitamins, metabolism of terpenoids and polyketides, metabolism of other amino acids, replication and repair, lipid metabolism and energy metabolism, these metabolic processes are necessary for the survival of animal These metabolic processes are essential for the survival, growth, and reproduction of the animal gut microbial community [49], and help yak to improve the utilization efficiency of fibrous grass and obtain more energy. Among them, energy is an essential component of feed that determines the feed intake of animals and plays an essential role in maintaining their growth, metabolism, and energy metabolism. In the present study, energy metabolism was higher in the C65 group than in the other two groups, suggesting that feeding a 65:35 concentrate-to-forage ratio diet contributes to the upregulation of the energy metabolism pathway in yaks. The current study indicates that different dietary concentrate ratios alter the putative gene function of yak fecal microbiota and that fecal microbiota under the feeding of 65:35 concentrate-to-forage ratio diets may improve yak metabolism and thus effectively promote yak growth.

## 5. Conclusions

The results of this study showed that DMI was higher in yaks fed a 65:35 concentrate-to-forage ratio diet, which contributed to the growth performance of yaks. We observed using 16S rRNA gene sequencing analysis that the alpha diversity of the bacterial community in the fecal microbiota and the relative abundance at the phylum and genus levels changed with the change in dietary concentrate-to-forage ratio, but there was no significant difference. Yaks fed a 65:35 concentrate-to-forage ratio were more capable of degrading forage cellulose and obtaining energy from indigestible polysaccharides and had a relatively lower risk of metabolic diseases, improving energy metabolism in yaks, which in turn effectively promoted yak growth. These observations provide new insights into the current understanding of the gut microbiota of yaks and provide evidence for the effect of dietary concentrate-to-forage ratio on the fecal microbiota.

## Figures and Tables

**Figure 1 animals-12-02334-f001:**
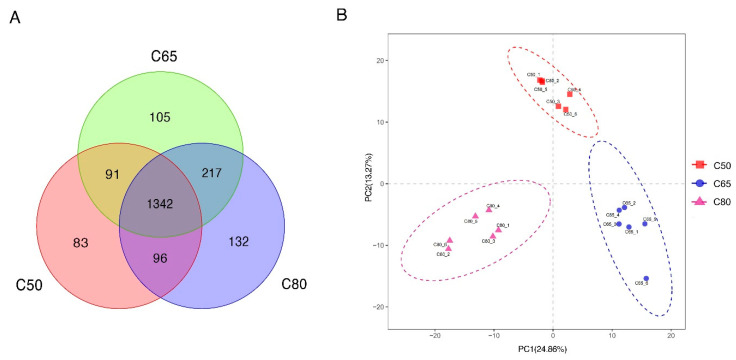
Differences of fecal bacterial community richness and OTUs in yak with concentrate-to-forage ration. Venn diagram (**A**) shows the three different kinds of fine forage than diet and similar OTUs. Principal coordinate analysis (PCoA) (**B**) of the fecal microbiota of yak with three concentrates to forage ratio diets.

**Figure 2 animals-12-02334-f002:**
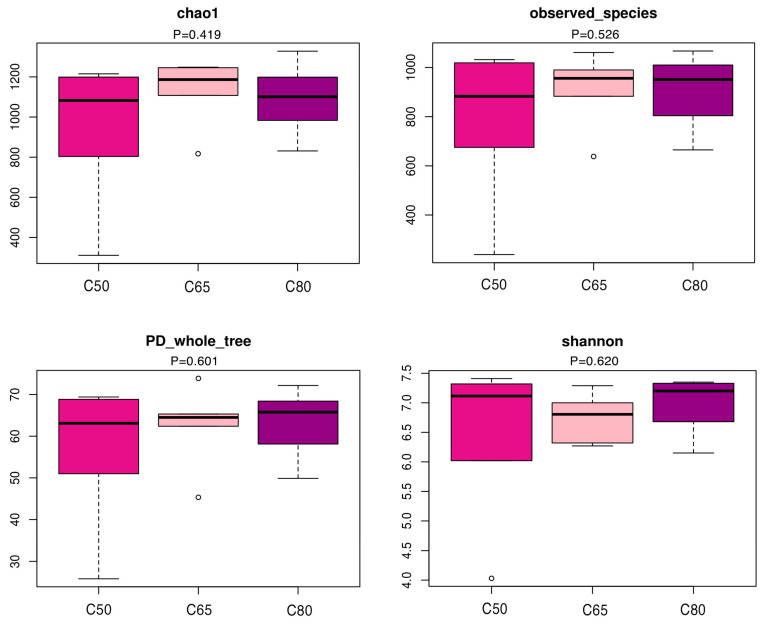
Effect of three diets of concentrate-to-forage ratio on the diversity of fecal microbiota of yaks. *p* < 0.05 was considered a significant difference.

**Figure 3 animals-12-02334-f003:**
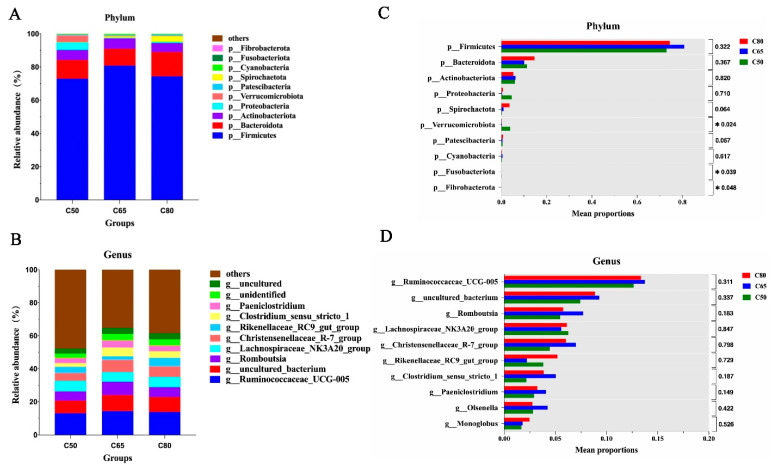
The bacterial composition of rumen samples from yaks with different dietary concentrate-to-forage ratios. Bacterial composition at the phylum (**A**) and genus (**B**) levels; Significantly different bacterial phylum (**C**) and genus (**D**) between groups, where different letters indicate significant differences between groups.

**Figure 4 animals-12-02334-f004:**
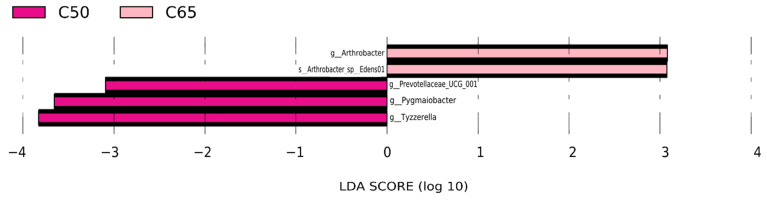
The LEfSe (Linear discriminant analysis Effect Size) profiles of microbial communities were compared between three different concentrates to forage ratios diets. Differences were found in the color of the most abundant taxa; purple: taxa abundant in C50, pink: taxa abundant in C65.

**Figure 5 animals-12-02334-f005:**
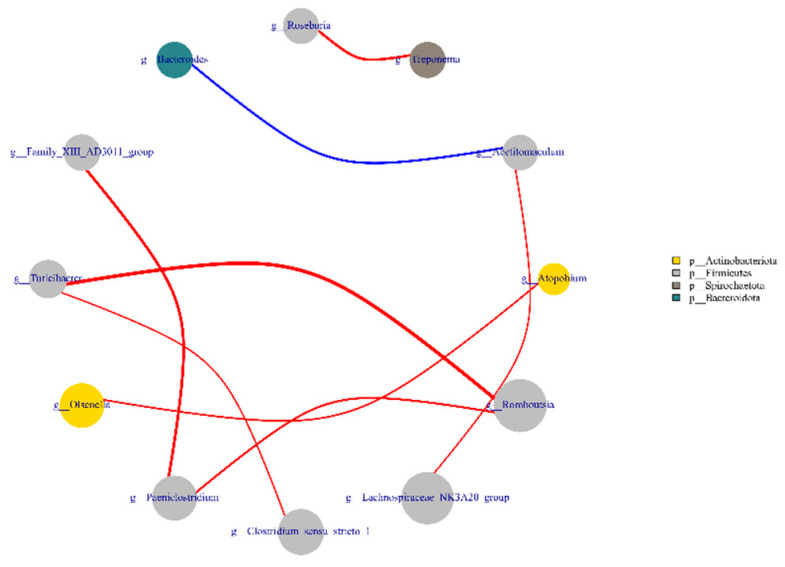
Analysis of co-occurrence network among microbial genera. Each co-occurrence pair between genus-level microbiota has an absolute Spearman rank correlation, with the red line indicating a positive correlation between two species and the blue line showing a negative correlation between two species, while the thickness of the line represents the magnitude of the correlation, with the thicker line representing a more robust correlation and the thinner line representing a weaker correlation.

**Table 1 animals-12-02334-t001:** Ingredients and nutritional composition of each diet (DM%).

Ingredients (%)	Group		
	C50	C65	C80
Oat hay	50.00	35.00	20.00
Corn	22.68	29.75	37.15
Wheat	6.25	8.39	10.42
Wheat bran	6.43	8.56	10.67
Rapeseed meal	6.36	8.55	10.61
Soybean meal	2.17	2.91	3.62
Palm oil powder ^(2)^	2.11	2.84	3.53
CaHPO4	1.00	1.00	1.00
NaCI	1.00	1.00	1.00
Premix ^(1)^	2.00	2.00	2.00
Nutrient composition (%)			
CP	12.41	13.18	13.87
ME MJ/kg	10.98	11.89	12.80
NDF	37.67	31.36	25.48
ADF	23.32	18.36	13.61
Ca	0.45	0.48	0.50
P	0.48	0.56	0.63

^(1)^ The premix provided the following per kg of the diet: VA 3 500 IU, VD 1 000 IU, VE 40 IU, Mn 40 mg, Fe 50 mg, Cu 10 mg, Zn 40 mg, Se 0.3 mg; ^(2)^ Others were measured values.

**Table 2 animals-12-02334-t002:** Effects of dietary Concentrate and Forage on growth performance of Yak.

Items ^1^	Group ^2^			SEM ^3^	*p*-Value
	C50	C65	C80		
Initial BW, kg	162.17	162.50	162.75	5.401	0.948
Final BW, kg	227.67	234.08	230.58	7.026	0.940
TWG, kg	65.50	75.58	67.83	3.956	0.581
ADG, kg/d	0.73	0.84	0.75	0.440	0.581
DMI, kg/d	5.84 ^b^	6.21 ^a^	5.83 ^b^	0.070	0.035
F/G	8.18	7.68	8.77	0.565	0.757

^1^ BW body weight, TWG The total weight, ADG average daily gain, DMI dry matter intake. F/G = DMI/ADG. ^2^ C50, diet contained 50% of concentrate; C65, diet contained 65% of concentrate; C80, diet contained 80% of concentrate.^3^ SEM, standard error of the mean.

**Table 3 animals-12-02334-t003:** Effect of different dietary concentrate ratios on the functional prediction of yak manure communities.

Item	Group ^a^			SEM ^b^	*p*-Value
	C50	C65	C80		
Carbohydrate metabolism	13.91	13.74	13.78	0.080	0.690
Amino acid metabolism	12.88	12.90	13.01	0.064	0.687
Metabolism of cofactors and vitamins	12.25	12.04	12.26	0.094	0.600
Metabolism of terpenoids and polyketides	10.41	10.66	10.69	0.136	0.688
Metabolism of other amino acids	6.82	6.71	6.75	0.052	0.674
Replication and repair	6.37	6.52	6.54	0.046	0.292
Lipid metabolism	5.47	5.69	5.68	0.071	0.390
Energy metabolism	5.35	5.50	5.47	0.046	0.412
Translation	3.60	3.64	3.68	0.031	0.520
Folding, sorting and degradation	3.46	3.50	3.51	0.016	0.439
Cell motility	3.13	3.37	3.36	0.115	0.651
Glycan biosynthesis and metabolism	3.38	3.02	3.24	0.106	0.394
Nucleotide metabolism	2.09	2.12	2.14	0.013	0.332
Biosynthesis of other secondary metabolites	2.02	1.94	2.02	0.017	0.074
Xenobiotics biodegradation and metabolism	2.38	2.15	1.41	0.212	0.143
Membrane transport	1.96	1.97	1.90	0.042	0.778
Cell growth and death	1.54	1.56	1.58	0.012	0.305
Transcription	1.26	1.29	1.31	0.022	0.705
Signal transduction	0.41	0.40	0.40	0.008	0.879

^a^ C50, diet contained 50% of concentrate; C65, diet contained 65% of concentrate; C80, diet contained 80% the concentrate. ^b^ SEM, standard error of the mean.

## Data Availability

The data that support the findings of this study are available from the corresponding author upon reasonable request, and the sequencing data are available from NCBI. The BioProject number is PRJNA864630.

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
