# Peer review of "Dynamics Changes of the Fecal Bacterial Community Fed Diets with Different Concentrate-to-Forage Ratios in Qinghai Yaks"

_animals, 2022, doi:10.3390/ani12182334_

Round 1

Reviewer 1 Report

Researchers evaluated the effect of different concentrations to coarse ratios on growth performance and fecal microorganisms of yaks by 16s 16S rRNA gene sequencing. As described by the author, this manuscript is the first to evaluate the effect of dietary concentrate to the coarse ratio in influencing fecal microorganisms in yaks. Interestingly, the better concentrate to forage ratio is 65:35 than 80: 20. This provides basic information for animal husbandry on the Tibetan Plateau. For the most part, the manuscript is well-written, and the results are clearly described, the discussion section of the manuscript is relatively full and has good readability, but for the sake of clarity of the manuscript. I have a few comments and suggestion.

Comments and Suggestions for Authors: 

1.       Line 23, Suggest deleting "housed", The general expression of fine to coarse ratio is housed

2.       Line 33, Is the initial weight inconsistent?

3.       Line 34, "F/G" the full name "DMI/ADG"

4.       Line 36, "significantly lower", Not significant, suggests deleting significant.

5.       Line 42, fecal microbiota Revise full text to unify fecal microbiota.

6.       Line 80-82, "pens". The expression is wrong, please correct it toTherefore, our study focused on the effect of the concentrate to forage ratio in the diet on the fecal microbiota of yaks.

7.       Line 84, The subject of the study is feces, please remove rumen-related words here.

8.       Line 98, "site" is modified to "located".

9.       Line 98, Please indicate whether the initial weight is consistent.

10.   Line 100, Please express clearly whether the diets are fed separately or together.

11.   Line 109-113, This experiment is a fine to coarse ratio housed feeding, so here is why there are grass samples collected for the grazing experiment.

12.   Line 125, A total of 6 heads were selected for the test.

13.   Line 179, Initial weight was the lowest in the C65 group, please verify that the expression was incorrect. Please check the initial weight data.

14.   Line 183,"FG" is modified to "F/G".

15.   Line 212,“fine to forage ratio Wrong expression, please correct

16.   Line249,Line255, There are only 2 groups in the figure, no three groups are seen.

Author Response

Dear reviewer:

We are very grateful for your efforts in our manuscript and for giving us the opportunity to resubmit a revised version of our manuscript. Those comments are all valuable and very helpful for revising and improving our paper, as well as the important guiding significance to our research. We have studied comments carefully and have made corrections which we hope meet with approval. Revised portions are marked in red on the paper. The main corrections in the paper and the responses to the reviewer’s comments are as flowing:

1.Line 23, Suggest deleting "housed", The general expression of fine to coarse ratio is housed

Response1: We appreciate these comments. It has been modified in the text.

2.Line 33, Is the initial weight inconsistent?

Response2: We express our deepest gratitude for your careful work and thoughtful suggestions. The data were checked and the initial weight of the animals was consistent.

3.Line 34, "F/G" the full name "DMI/ADG"

Response3: We appreciate these comments. It has been revised and marked in red.

4.Line 36, "significantly lower", Not significant, suggests deleting significant.

Response4: We express our deepest gratitude for your careful work and thoughtful suggestions. It has been revised and marked in red.

5.Line 42, “fecal microbiota” Revise full text to unify fecal microbiota.

Response5: We express our deepest gratitude for your careful work and thoughtful suggestions. The fecal microbiota has been harmonized in the full text revision and marked in red. (Line63,65,79,81,87,209,306,310,328,329,365,367,380)

6.Line 80-82, "pens". The expression is wrong, please correct it to“Therefore, our study focused on the effect of the concentrate to forage ratio in the diet on the fecal microbiota of yaks.”

Response6: We are grateful for the comments. It has been revised and marked in red. (Line67-68)

7.Line 84, The subject of the study is feces, please remove rumen-related words here.

Response7: Thank you very much for your suggestion. It has been revised and marked in red.

8.Line 98, "site" is modified to "located".

Response8: We thank you for your questions about this study. It has been revised and marked in red. (Line96)

9.Line 98, Please indicate whether the initial weight is consistent.

Response9: Thank you very much for your suggestion. We apologize that the data were verified and reviewed for inconsistencies in initial weight due to writing errors, which have been corrected in the text and marked in red. (Line96)

10.Line 100, Please express clearly whether the diets are fed separately or together.

Response10: Thank you very much for your suggestion. The diets were all fed individually and have been modified and marked in red in the text. (Line98)
11.Line 109-113, This experiment is a fine to coarse ratio housed feeding, so here is why there are grass samples collected for the grazing experiment.

Response11: Thank you very much for your suggestion. I am sorry, but the yaks are selected from grazing pastures, and some of the pasture grasses are also taken, but the text is not related to grazing pastures, here the part is still deleted.

12.Line 125, A total of 6 heads were selected for the test.

Response12: Thank you very much for your suggestion. I apologize for the lack of clarity here, and have corrected it in the text and marked it in red. From the 36 yaks, 6 yaks were randomly selected from each of the three groups, and a total of 18 yaks were selected. (Line1222-123)
13.Line 179, Initial weight was the lowest in the C65 group, please verify that the expression was incorrect. Please check the initial weight data.

Response13: Thank you very much for your suggestion. We apologize that the data were verified and reviewed for inconsistencies in initial weight due to writing errors, which have been corrected in the text .
14.Line 183,"FG" is modified to "F/G".

Response14: We thank you for your questions about this study. Changes have been made in the text and marked in red. (Line181)
15.Line 212,“fine to forage ratio” Wrong expression, please correct

Response15: Thank you very much for your suggestion. The text has been changed to "concentrate to forage ratio" and is marked in red. (Line209)
16.Line249,Line255, There are only 2 groups in the figure, no three groups are seen.

Response16: We express our deepest gratitude for your careful work and thoughtful suggestions. The LEfSe analysis was indeed between the three groups, as only species that differed between the two groups were shown at the set value of LDA>3.

Our deepest gratitude goes to you for your careful work and thoughtful suggestions that have helped improve this paper substantially.

Reviewer 2 Report

This study describes the distribution of yak fecal microbiota under different diets, which in most cases may provide a reference for livestock husbandry in the Tibetan plateau. The manuscript discusses that under the conditions of this experiment, yaks were best fattened with a 65:35 ratio of forage concentrate, and the results conclusions are clearly described, however, the clarity of the manuscript should be improved, the author did not explain why using the concentrate ratio of C50, C65, and C80 for experimental design, is there any scientific basis for reference. In addition, some details are commented on as follows.

1.      Page 1. Line 33, Please indicate whether the initial weight of the animals is consistent.

2.      Page 1. Line 36, No significance was found in the results, and significance is expressed here, please verify by the authors.

3.      Page 3. Line 92, In the "material and methods" part, are the experimental animals separated for feeding in each group? How long does the experiment last? Please show detailed information.

4.      Page 3. Line 100, Please express clearly whether the diets are fed separately or together.

5.      Page 3. Line 109, In the "material and methods" part, this experiment was done under-housed conditions, why did the authors write that the forage mix samples were collected from the grazing playground?

6.      Page 3. Line 125, In the "material and methods" part, the author sampled 36 samples, why there were only 18 fecal fluid samples for gene sequencing.

7.      Page 8. Line 249, In the " results " part, the LEfSe analysis only showed 2 groups, so why did the authors say that they showed significant differences between the three groups of fine to coarse ratios?

Author Response

Dear reviewer:

We are very grateful for your efforts in our manuscript and for giving us the opportunity to resubmit a revised version of our manuscript. Those comments are all valuable and very helpful for revising and improving our paper, as well as the important guiding significance to our research. We have studied comments carefully and have made corrections which we hope meet with approval. Revised portions are marked in blue on the paper. The main corrections in the paper and the responses to the reviewer’s comments are as flowing:

This study describes the distribution of yak fecal microbiota under different diets, which in most cases may provide a reference for livestock husbandry in the Tibetan plateau. The manuscript discusses that under the conditions of this experiment, yaks were best fattened with a 65:35 ratio of forage concentrate, and the results conclusions are clearly described, however, the clarity of the manuscript should be improved, the author did not explain why using the concentrate ratio of C50, C65, and C80 for experimental design, is there any scientific basis for reference. In addition, some details are commented on as follows.

Response:We express our deepest gratitude for your careful work and thoughtful suggestions. According to our survey, the current level of concentrate used on farms is around 50%-80%. Combined with other articles on the concentrate to forage ratio, high concentrate ration levels are generally 70-80%, but concentrate levels below 50 % do not promote production either. Therefore, we chose between 50-80 % and set three shaving levels of concentrate to forage ratio, 50.65.80, to conduct a study to find the right concentrate to forage ratio on yaks and to promote production.

References:1.Jiang, Y., Dai, P., Dai, Q., Ma, J., Wang, Z., Hu, R., Zou, H., Peng, Q., Wang, L., Xue, B. (2021). Effects of the higher concentrate ratio on the production performance, ruminal fermentation, and morphological structure in male cattle-yaks. Vet Med Sci. 8(2):771-780. doi: 10.1002/vms3.678.

2.Zhang, J., Shi, H. T., Wang, Y. C., Li, S. L., Cao, Z. J., Yang, H. J., et al. (2020). Carbohydrate and amino acid metabolism and oxidative status in holstein heifers precision-fed diets with different forage to concentrate ratios. Animal. 14(11), 2315-2325. doi: 10.1017/S1751731120001287

  1. Page 1. Line 33,Please indicate whether the initial weight of the animals is consistent.

Response1: We appreciate these comments. The data were checked and the initial weight of the animals was consistent.

  1. Page 1. Line 36,No significance was found in the results, and significance is expressed here, please verify by the authors.

Response2: We express our deepest gratitude for your careful work and thoughtful suggestions. The p-value for DMI is 0.035, so it is expressed here as significant.

  1. Page 3. Line 92,In the "material and methods" part, are the experimental animals separated for feeding in each group? How long does the experiment last? Please show detailed information.

Response3: We appreciate these comments. The experimental animals were fed separately. The pretest period was 15 d, and the normal test period was 90 d. Detailed information has been added to the text and is marked in blue. (Line96,102-103)

  1. Page 3. Line 100,Please express clearly whether the diets are fed separately or together.

Response4: We express our deepest gratitude for your careful work and thoughtful suggestions. Diets were fed separately and have been added and marked in the text. (Line98)

  1. Page 3. Line 109,In the "material and methods" part, this experiment was done under-housed conditions, why did the authors write that the forage mix samples were collected from the grazing playground?

Response5: We express our deepest gratitude for your careful work and thoughtful suggestions. I am sorry, but the yaks are selected from grazing pastures, and some of the pasture grasses are also taken, but the text is not related to grazing pastures, here the part is still deleted. (Line109-110)

  1. Page 3. Line 125,In the "material and methods" part, the author sampled 36 samples, why there were only 18 fecal fluid samples for gene sequencing.

Response6: We are grateful for the comments. In this experiment, 36 yaks were fed and 36 yaks were collected, and six fecal samples from each group were selected for gene sequencing.

  1. Page 8. Line 249,In the " results " part, the LEfSe analysis only showed 2 groups, so why did the authors say that they showed significant differences between the three groups of fine to coarse ratios?

Response7: Thank you very much for your suggestion. The LEfSe analysis was indeed between the three groups, as only species that differed between the two groups were shown at the set value of LDA>3.

Our deepest gratitude goes to you for your careful work and thoughtful suggestions that have helped improve this paper substantially.

Reviewer 3 Report

All my comments ae reported in the joined file.

Author Response

Dear reviewer:

We are very grateful for your efforts in our manuscript and for giving us the opportunity to resubmit a revised version of our manuscript. Those comments are all valuable and very helpful for revising and improving our paper, as well as the important guiding significance to our research. We have studied comments carefully and have made corrections which we hope meet with approval. Revised portions are marked in green on the paper. The main corrections in the paper and the responses to the reviewer’s comments are as flowing:

line 18,19 : Why do you consider that fecal microbiota collection is the same than rumen microbiota collection ?

Response1: I apologize for the unclear expression here, I think the fecal microbiota and rumen microbiota must be different, the meaning I wanted to express here is that collecting feces has no adverse effect on yaks, after review, the expression here is considered unsatisfactory and has been deleted.

line 37,38 : what is significant and what is not, using which test ?

Response2: We express our deepest gratitude for your careful work and thoughtful suggestions. Statistical analyses were performed using SPSS 24.0 (SPSS Inc., Chicago, IL, USA). The statistical difference between normally distributed data was analyzed using a One-Way ANOVA. p-values< 0.05 indicated significant differences,p-values > 0.05 indicated no significant differences. I apologize that here, due to a writing error, I have written significant results that were not significant and have made changes in the text.

line 54-56 : you should reformulate

Response3: We appreciate these comments. We have made careful changes here and marked it in green. (Line 52-56)

line 64 : why is it important to understand changes in fecal microbiota rather than in rumen microbiota?

Response4: We express our deepest gratitude for your careful work and thoughtful suggestions. Most of the gastrointestinal flora of mammals is concentrated in the hindgut, mainly in the colon. When comparing the structure of fecal flora and colon content flora, it is found that fecal flora can represent the intestinal content flora. This section has been added to the manuscript and is marked in green. (Line 75-78)

line 70 : Because the rumen is a fermentation chamber (anaerobic place), I think it is not the same to collect feces microbiota and rumen microbiota (and it is not the same stage of digest).

I am not convinced that you can justify to collect fecal microbiota rather than rumen microbiota only because of physiological damage especially because studies on rumen microbiota exists and was ethically approved. I think you are looking for something which is different (I would be surprised that you find the same microbes than in rumen) and should assume that.

Response5: We express our deepest gratitude for your careful work and thoughtful suggestions. We very much agree with your point of view. What we want to express here is that the collection of feces is harmless to yaks. After careful consideration, we think it is more appropriate to delete this part, so we have done the deletion.

Line 75-77 : I think this is the main reason for your study and should be mentioned since the beginning

Response6: We are grateful for the comments. The content of this section has been adjusted from the beginning. ( Line 57-81)

line 156 : Why did you do OTUs rather than ASV ?

Ideally, I would suggest you to obtain ASVs and perform your analyses on ASVs to improve the quality of your analyses.

Response7: We express our deepest gratitude for your careful work and thoughtful suggestions. Having referred to many studies, it was found that some of them used OTU analysis, so OTU analysis was also chosen for this study.

The references are as follows.

  1. Ma L, Xu S, Liu H, Xu T, Hu L, Zhao N, Han X, Zhang X. Yak rumen microbial diversity at different forage growth stages of an alpine meadow on the Qinghai-Tibet Plateau. PeerJ. 2019 Sep 19;7:e7645. doi: 10.7717/peerj.7645.
  2. Cui K, Qi M, Wang S, Diao Q, Zhang N. Dietary energy and protein levels influenced the growth performance, ruminal morphology and fermentation and microbial diversity of lambs. Sci Rep. 2019 Nov 12;9(1):16612. doi: 10.1038/s41598-019-53279-y.
  3. Wang Q, Wang Y, Wang X, Dai C, Tang W, Li J, Huang P, Li Y, Ding X, Huang J, Hussain T, Yang H, Zhu M. Effects of dietary energy levels on rumen fermentation, microbiota, and gastrointestinal morphology in growing ewes. Food Sci Nutr. 2020 Nov 10;8(12):6621-6632. doi: 10.1002/fsn3.1955.
  4. Han L, Xue W, Cao H, Chen X, Qi F, Ma T, Tu Y, Diao Q, Zhang C, Cui K. Comparison of Rumen Fermentation Parameters and Microbiota of Yaks From Different Altitude Regions in Tibet, China. Front Microbiol. 2022 Feb 10;12:807512. doi: 10.3389/fmicb.2021.807512. 
  5. Hou L, Wang L, Qiu Y, Xiong Y, Xiao H, Yi H, Wen X, Lin Z, Wang Z, Yang X, Jiang Z. Effects of Protein Restriction and Subsequent Realimentation on Body Composition, Gut Microbiota and Metabolite Profiles in Weaned Piglets. Animals (Basel). 2021 Mar 4;11(3):686. doi: 10.3390/ani11030686.

Line 180- 182 : you deal with non significant result (cf Table2). Moreover there are some inconsistencies.

« In addition, DMI and F/G in the 181 C65 group were significantly higher than in the other two groups (p > 0.05) » the pvalue mentioned and what you wrote just before are inconsistent. Reading table 2, the only significant variable is DMI.

Response8: We express our deepest gratitude for your careful work and thoughtful suggestions. I am very sorry for the wrong expression here due to a writing error, which has been modified into " In addition, DMI in the C65 group was significantly higher than in the other two groups (p < 0.05). The F/D of the C65 group was lower than that of the other two groups." and marked in green. ( Line 177-179)

Line 189 : I don't understand. It was not fecal microbiota ? Don't you have 36 yaks samples ? I don'tunderstand why do you have 18 samples.

Response9: We express our deepest gratitude for your careful work and thoughtful suggestions and questions. I apologize for this. Due to a writing error, " rumen fluid " was written here and has been corrected to " yak fecal " and marked in green. In this experiment, 36 yaks were fed and 36 yaks were collected, and six fecal samples from each group were selected for gene sequencing. Therefore, only 18 fecal samples were used for gene sequencing. (Line 186)

Line 200 : « concentratios »

Response10: We appreciate these comments. The word " concentratios " has been modified to " concentrate" in the text and is marked in green. ( Line 195,196)

line 201- 208 : Figure 2 does not show any significant differences. What does allow you to say that ?May you remind here the test you performed for the significance ?

Response11: We express our deepest gratitude for your careful work and thoughtful suggestions and questions. Although Figure 2 is not significant, there are differences in specific values, which can also indicate that the fecal microbiota of the C65 group has higher richness but lower diversity. The flora richness of the C50 group was low, while the flora diversity of the C80 group was high. These results indicate that the α-diversity of the bacterial community in fecal microbiota changes with the change of dietary concentrate to crude ratio. Statistical analyses were performed using SPSS 24.0 (SPSS Inc., Chicago, IL, USA). The statistical difference between normally distributed data was analyzed using a One-Way ANOVA. p-values< 0.05 indicated significant differences,p-values > 0.05 indicated no significant differences. The data were first analyzed for chi-square, and a p-value greater than 0.05 indicated that the variance of the groups was chi-square, and a one-way ANOVA could be used to compare the differences; if the p-value was less than 0.05, the variance of the groups was not chi-square, and a non-parametric test was used to compare the differences between the groups.

Line 222 : « The other fewer doors are Verrucomicrobiota (1.47%), »

what do you mean exactly ?

Response12: We appreciate these comments. I am very sorry that there is an error in expression here and have corrected the " doors " to " accounted for phylum " and marked it in green in the text. ( Line 217)

Figure 3 : axis labels are too small. Also check the quality (600dpi)

Which test did you perform to determine if results are significant ?

Response13: We express our deepest gratitude for your careful work and thoughtful suggestions and questions. Statistical analyses were performed using SPSS 24.0 (SPSS Inc., Chicago, IL, USA). The statistical difference between normally distributed data was analyzed using a One-Way ANOVA. p-values< 0.05 indicated significant differences,p-values > 0.05 indicated no significant differences. The data were first analyzed for chi-square, and a p-value greater than 0.05 indicated that the variance of the groups was chi-square, and a one-way ANOVA could be used to compare the differences; if the p-value was less than 0.05, the variance of the groups was not chi-square, and a non-parametric test was used to compare the differences between the groups.

Figure 4 : we can't read which taxa make a difference between C50 and C65. It is too small.

Response14: We express our deepest gratitude for your careful work and thoughtful suggestions. LEfSe showed species with LDA scores greater than 3 that differed. Show the species with significant differences in abundance in different groups.  The most abundant bacterial genera in the C50 group were Prevotellaceae_UCG_001, Pygmaiobacter and Tyzzerella, while Arthrobacter and Ar-throbacter_sp_Edens01 were C65 groups were more abundant.

Line 262 : 0.008 is a really weak correlation. Is it really relevant to notice it ? For me there is no correlation here.

Response15: We are very grateful to you for your advice. After careful review, it is confirmed that the correlation here is weak and has been deleted in the text.

Line 294 & 390: about ADG, it is not significant (cf Table 2)

Response16: We express our deepest gratitude for your careful work and thoughtful suggestions. The implication here is, ADG, although not significant, was higher in the 65 group in terms of specific values.

line 392- 394 : I don't see any significant results on diversity, in your article to conclude that (cf Figure2)

Response17: We are very grateful to you for your advice. Although the diversity data in the article are less significant, there are differences between the three groups in terms of specific values. The different diets with different concentrate to coarse ratios affected the structure of the yak fecal microbiota, but the differences were not significant. The conclusion that diversity was not significant in this section has been removed.

Our deepest gratitude goes to you for your careful work and thoughtful suggestions that have helped improve this paper substantially.

Round 2

Reviewer 3 Report

I use purple to answer you on your own answer.

Author Response

Dear Reviewers:
We are very grateful for your efforts on our manuscript and for giving us the opportunity to resubmit a revised version of our manuscript. The comments were valuable and very helpful in revising and improving our paper, as well as providing important guidance for our research. We have carefully studied these comments and made revisions, which we hope will be approved by you. The revised parts and the responses are marked in yellow on the paper.
